# Experimental Models for Testing the Efficacy of Pharmacological Treatments for Neonatal Hypoxic-Ischemic Encephalopathy

**DOI:** 10.3390/biomedicines10050937

**Published:** 2022-04-19

**Authors:** Elisa Landucci, Domenico E. Pellegrini-Giampietro, Fabrizio Facchinetti

**Affiliations:** 1Department of Health Sciences, Section of Clinical Pharmacology and Oncology, University of Florence, 50139 Florence, Italy; elisa.landucci@unifi.it; 2Department of Experimental Pharmacology and Translational Science, Corporate Pre-Clinical R&D, Chiesi Farmaceutici S.p.A., 43122 Parma, Italy; f.facchinetti@chiesi.com

**Keywords:** cerebral ischemia, hypoxia, oxygen and glucose deprivation, cell cultures, organotypic hippocampal slices, neonate animal models, efficacy studies, toxicological studies, drug development

## Abstract

Representing an important cause of long–term disability, term neonatal hypoxic-ischemic encephalopathy (HIE) urgently needs further research aimed at repurposing existing drug as well as developing new therapeutics. Since various experimental in vitro and in vivo models of HIE have been developed with distinct characteristics, it becomes important to select the appropriate preclinical screening cascade for testing the efficacy of novel pharmacological treatments. As therapeutic hypothermia is already a routine therapy for neonatal encephalopathy, it is essential that hypothermia be administered to the experimental model selected to allow translational testing of novel or repurposed drugs on top of the standard of care. Moreover, a translational approach requires that therapeutic interventions must be initiated after the induction of the insult, and the time window for intervention should be evaluated to translate to real world clinical practice. Hippocampal organotypic slice cultures, in particular, are an invaluable intermediate between simpler cell lines and in vivo models, as they largely maintain structural complexity of the original tissue and can be subjected to transient oxygen–glucose deprivation (OGD) and subsequent reoxygenation to simulate ischemic neuronal injury and reperfusion. Progressing to in vivo models, generally, rodent (mouse and rat) models could offer more flexibility and be more cost-effective for testing the efficacy of pharmacological agents with a dose–response approach. Large animal models, including piglets, sheep, and non-human primates, may be utilized as a third step for more focused and accurate translational studies, including also pharmacokinetic and safety pharmacology assessments. Thus, a preclinical proof of concept of efficacy of an emerging pharmacological treatment should be obtained firstly in vitro, including organotypic models, and, subsequently, in at least two different animal models, also in combination with hypothermia, before initiating clinical trials.

## 1. Introduction

Perinatal hypoxic–ischemic encephalopathy (HIE), which affects 2–4/1000 full term births, is a major cause of acute mortality and chronic neurological morbidity [1], being frequently associated with neurocognitive impairment, cerebral palsy, and seizure disorders [2]. Although therapeutic hypothermia is becoming standard clinical care for moderate to severe neonatal encephalopathy, it is only partially effective, with almost 50% of treated infants having adverse outcomes [3]. Current hypothermia protocols have consistently involved starting treatment within the first 6 h of life, with systemic cooling to either 34.5 ± 0.5 °C for head cooling or 33.5 ± 0.5 °C for whole-body cooling, and continuing treatment for 48–72 h [4]. However, many newborns with encephalopathy have been exposed to hypoxia and/or ischemia well before birth, with a consequent delay of hypothermia delivery before clinical diagnosis. Novel therapeutic strategies capable of augmenting neuroprotection and/or neurodegeneration in combination with therapeutic hypothermia are highly needed to reduce the neurological consequences of HIE [1].

Hypoxia–ischemia of sufficient severity to deplete cerebral energy reserves activates multiple pathways leading to ongoing cell death, including excitotoxicity, oxidative stress, and mitochondrial stress, apoptosis, and microglial activation [5]. Many substances possessing antioxidative properties show the ability to protect the neonatal brain from hypoxia–ischemia in experimental models [6], such as N-acetylcysteine [7] and mitochondria-targeted antioxidants [8]. Even though there is a partial restoration of cerebral energy stores in the early period of reperfusion, in both asphyxiated infants and pre-clinical animal models, a secondary energy failure occurs during the subsequent hours to days [9]. Clinically, the severity of the second energy failure as determined with ^31^P-MRS has been shown to correlate with adverse neurological outcomes at 1 and 4 years of follow-up [10,11]. The tertiary phase can last for months/years and result in developmental disturbances and has been associated with chronic neuroinflammation, loss of trophic support, and impaired connectivity and maturation [12], thus suggesting a therapeutic window of intervention up to several days after the insult.

Although it is important for an experimental drug for HIE to be thoroughly evaluated for its mechanism of action and neuroprotective efficacy in in vitro cell-based and organotypic models mimicking hypoxic–ischemic/excitotoxic conditions, the use of pre-clinical animal models continues to be essential for supporting full development. Commonly, experimental models for the study of neonatal encephalopathy in term infants involve induction of hypoxic–ischemic or focal ischemic lesions in various species, including rodents and piglets [9]. Such pre-clinical models were instrumental for demonstrating that mild, induced hypothermia can improve neurological recovery after global moderate to severe hypoxia–ischemia [13]. Since therapeutic hypothermia has become a standard of care therapy for neonatal encephalopathy, translational preclinical testing of therapeutic agents should occur also in comparison as well as in combination with therapeutic hypothermia [14], and therapeutic interventions must be initiated after the completion of the insult to better mimic the clinical condition [15].

## 2. In Vitro Models

A number of in vitro and ex vivo models [3] have been developed in the past few decades to study the mechanisms of cerebral ischemia and hypoxia and to predict the effects of drugs before they can be used at a larger scale and in more complex systems in vivo. As compared to the latter, in vitro models have the obvious advantage of being less expensive and shorter in duration, besides for sparing animal lives, thanks to the use of immortalized cell lines and to the large number of primary culture samples than can be prepared from one single animal.

Due to the complex nature of HIE, it is quite difficult to successfully model the disease with a single in vitro system. However, these preparations allow the investigation of the biochemical and molecular mechanisms that lead to post-ischemic/hypoxic damage and the screening and accurate pharmacological characterization of candidate neuroprotective compounds. Hypoxic and ischemic-like conditions in vitro can be reproduced by blocking the cellular metabolism with 2-deoxyglucose or sodium azide or by inducing excitotoxicity with NMDA or AMPA receptor agonists. However, the most frequently used method to simulate an HIE or stroke-like environment is to remove the supply of oxygen and glucose to the cells, a method known as oxygen–glucose deprivation (OGD) [16], which is very often followed by a return to a normoxic and normoglycemic condition that mimics reperfusion.

### 2.1. Dissociated Neuronal and Mixed Glial/Neuronal Cultures

As compared to immortalized cell lines such like the human teratoma-derived NT2 cell line or SH-SY5Y dopaminergic cells derived from bone marrow neuroblastoma, primary neuronal cultures have undoubtedly the advantage of being composed of almost homogeneous normal neuronal cells (the glial component always being <20% and often <5%). However, as compared to more complex systems, this homogeneity is a limitation in that this does not allow one to keep into account the profound reciprocal interactions between neurons and glial cells. As a result, their response to insults like hypoxia and ischemia is often unpredictable and very different to what typically occurs in in vivo models or in humans, requiring sometimes very prolonged periods of OGD (even 2 to 4 h) to produce neuronal damage that is evident in vivo after few minutes. Sometimes other types of brain cells, such as astrocytes [17], microglia [18], or human brain microvascular endothelial cells (hBMVECs) [19] are used to generate OGD models, but their response is even more distant to what is observed in the brain of animals or infants with HIE.

An important improvement in the physiological relevance of in vitro HIE models is the development and use of co-culture models, which are based on diverse brain cell populations, often cultured in repeated monolayers, permitting the exchange of substrates and signaling molecules among various cell types, as it occurs in original cerebral tissue in vivo. Goldberg and Choi [20] were the first to develop a model of mixed cortical mouse in which neurons were cultured on top of a layer of astrocytes and exposed to OGD for various time intervals that produced specific differences in vulnerability and injury severity between neurons and glial cells. Typically, an OGD of 1 h in this system produces a selective degeneration of neurons, as detected by measuring the release of LDH release or by morphological approaches and leaves the glial cell layer unaffected. This model can be easily adapted to various experimental conditions in order to replicate diverse cellular aspects in response to hypoxia or ischemia in a reproducible manner. Many studies, which can be classified as mechanistic studies, have thoroughly used mixed cortical cells exposed to OGD to evaluate the role of receptors, channels, and intracellular genes and proteins in the pathogenesis of cerebral ischemia, while a parallel set of studies has examined the neuroprotective efficacy of various interventions, including natural or synthetic chemical entities, to be proposed as therapeutic options for cerebral hypoxic–ischemic disorders.

### 2.2. D Brain Organoids

An influential class of supracellular models is represented by pluripotent stem cells, which show a remarkable ability to self-organize and differentiate in vitro in three-dimensional aggregates, known as organoids or organ spheroids, recapitulating aspects of human brain development and function. Region-specific 3D brain cultures can be assembled to model complex cell–cell interactions and to generate circuits in human brain assembloids. Brain region-specific spheroids can undergo different maturation stages and can mimic immature brain conditions and maturation [21]. Human 3D brain-region-specific organoids can be utilized to study the effect of hypoxic insults utilizing a variety of read outs, including live-cell imaging, calcium dynamics, electrophysiology, cell purification, single-cell transcriptomics, and immunohistochemistry studies. Despite the challenges posed by the heterogeneous nature of brain organoids generated from pluripotent stem cells, such a human cell-based 3D model is potentially amenable to drug screening and to mechanistically studying neuroprotective therapies. [22].

### 2.3. Organotypic Hippocampal Slices

Organotypic hippocampal slices are an invaluable intermediate between simpler cell cultures and in vivo models, as they largely maintain structural complexity, synaptic organization, and receptor expression of the original tissue [23]. In addition, organotypic slice cultures preserve the interaction between neurons and glial cells, necessary for supporting the energetic status of neurons under ischemic conditions, and retain synaptic plasticity mechanisms and a susceptibility to damaging injuries that is similar to what occurs in vivo, even for what concerns the selectivity of the vulnerable region (i.e., CA1 after ischemic-like injury and CA3 following incubation with kainic acid [24]). Organotypic hippocampal slices are usually prepared from 7 to 9 day-old rats and can be cultured in vitro for 2 or 10 days in vitro, producing both “immature” and “mature” slices, respectively [25], and making them ideal experimental candidates for the study of developing age diseases like HIE.

Hippocampal slice cultures can be studied with biochemical, molecular biological, electrophysiological, and morphological approaches in order to monitor adaptations of neuronal circuits in response to long-term incubation with drugs, and hence they can be utilized as screening platforms to identify the pharmacological properties of novel therapeutics. In particular, hippocampal slices can be subjected to transient oxygen–glucose deprivation (OGD) and subsequent reoxygenation to simulate ischemic neuronal injury and reperfusion [26,27]. Moreover, hippocampal slice cultures exposed to OGD respond to therapeutic hypothermia, thus allowing the testing of neuroprotective agents in combination with hypothermia [28,29]. Assessing the range of neuroprotective concentrations and the time window of effective administration of a given agent in vitro, in conjunction with pharmacokinetic determination of plasma and cerebral spinal fluid levels in vivo, would be essential for determining the range of doses to be utilized in in vivo experimental models to be ultimately translated in patients.

## 3. At Term and Juvenile In Vivo Animal Models

### 3.1. The Rice–Vannucci Model

The classic Rice–Vannucci HIE model (in which 7-day-old rat pups undergo unilateral ligation of the common carotid artery followed by exposure to 8% oxygen hypoxic air) is a model of neonatal stroke that has greatly contributed to the research in this field [30]. The model has been adapted also to mice to exploit strains genetically modified for mechanistic studies and validation of potential therapeutic targets [31]. The Rice–Vannucci model of HIE is well developed and widely applied with different periods of hypothermia used as a neuroprotective strategy in combination with other agents [9]. However, varying degrees of neuroprotection with different times of initiation and durations of hypothermia following HI have been described [32]. So far, most rodent data are obtained using exposure to 5–6 h of therapeutic hypothermia (usually between 32 and 33 °C).

The rodent HIE model is the most convenient, cost-effective, and widely used animal model [9]. Rodent pups do not require intensive care after the induction of the lesion, and follow up on neurological and functional parameters and/or lesion evolution over time is possible and responds to therapeutic hypothermia [33]. However, major differences between rodent (lissencephalic) and human (gyrencephalic) brain structure and maturation exist that hamper full translational applicability of the results obtain in rodents to humans. The great variability of infarct volumes among littermates concomitantly subjected to the identical hypoxic–ischemic insult requires an adequate sample size in the experimental groups. Responsiveness to neuroprotective treatments is highly dependent on the extent of the severity of the ischemia-induced brain lesion [34] and can be sex-dependent [31]. Importantly, the large injured areas observed in the rodent HIE model do not necessarily translate into severe neurological deficits as would probably occur in humans [35].

### 3.2. Piglet Model of Neonatal Encephalopathy

Newborn piglet brain shows greater similarities than rodents to human brain structure (gyrencephalic) and maturation. The piglet model is amenable to intravenous infusions and drug therapeutics on top of hypothermia to assess additive effects of new therapies in a translational mode [15]. Piglet body surface area and physiology are more comparable to human neonates, allowing more accurate pharmacokinetics and human dose predictions [15]. Real time monitoring of physiological parameters can also allow for accurate safety pharmacology studies [15], and several translational read outs have been described showing that the model mimics many aspects of the human condition [10].

Experiments in the piglet model can be conducted up to 48–72 h post-lesion, so no follow-up is possible on neurological and functional parameters and/or lesion evolution over time. The lack of homogenous genetic background between individuals can enhance variability. Piglet studies require an intensive care unit and dedicated and trained personnel during the whole duration of the in vivo experimental procedures. High costs are associated with lengthy timelines, as usually only one or two animals can be processed in one experimental session, thus limiting the number of subjects that can be included in a study. High mortality and differences between protocols result in variability in the extent of brain injury and make it difficult to standardize the model and make firm comparisons between laboratories. A real time titration of the HI insult is recommended to reduce variability [15].

## 4. Near-Term Animal Models

Non-human primate models of acute perinatal asphyxia using umbilical cord occlusion (UCO) prior to delivery to produce moderate-to-severe HIE have been described [36,37]. The animals develop severe asphyxia associated with cerebral palsy-like motor abnormalities [36,37]. However, ethical concerns, costs, and the complexity of conducting non-human primate studies limit the availability of this model for drug discovery.

Sheep studies have been performed during pregnancy to correlate to relevant maturation stages in the human [38]. In preterm sheep models, fetuses are more prone to white and deep gray matter injury, with an increased vulnerability of cortical gray matter with advancing gestation. Cerebral ischemia models in fetal sheep, induced by bilateral transient occlusion of the carotid arteries, were first developed in the near-term fetus and, later, during mid gestation [39]. The chronically instrumented fetal sheep umbilical cord occlusion model is global and allows for examination of intrauterine pathophysiology and the contribution of other organs on the brain, without the influence of anesthesia [33]. However, fetal models are complicated by maternal/placenta metabolism, which is not present in the human term HIE. Another point of consideration of this model is severe operational trauma and long recovery time of pregnant sheep. The sheep model is hardly cost effective, considering the technical complexity related to conducting such studies, thus limiting the availability of this model for drug discovery.

In rabbits, intrauterine ischemia is induced at 92% gestation (E29) to mimic at-term injury. MRI observation of deep brain injury 6 to 24 h after near-term hypoxia–ischemia predicts dystonic hypertonia postnatally [40]. Interestingly, in this model, motor deficits in rabbits progress from initial hypotonia to hypertonia, similar to human cerebral palsy, but in a compressed timeframe [41,42]. Although of more limited accessibility, the rabbit model could complement rodent studies in drug discovery, contributing also with important information on motor deficits.

In general, in near-term models, pharmacokinetics and pharmacological studies are complicated by the involvement of the maternal/placenta system, which affects the metabolism, absorption, distribution, and excretion of the drug.

## 5. The Ideal Timeline and Examples of Practical Approaches

### 5.1. Erythropoietin

Among the examples that can be mentioned as an ideal practical approach from bench to bedside in testing the efficacy of therapeutic agents for neonatal HIE, the story of erythropoietin is probably one of the most compelling. This cytokine is well known for being a crucial hematopoietic growth factor but also for its role in the proliferation and differentiation of neurons, glia, and endothelial cells, both during brain development and following hypoxia. Initial in vitro studies demonstrated that exposure to erythropoietin during and for 24 h after the insult was neuroprotective against OGD and NMDA toxicity in organotypic hippocampal slices [43] and, quite interestingly, that erythropoietin could also induce ischemic tolerance when added to the incubation medium of neuronal cortical cultures 24 h prior to OGD exposure [44]. In vivo neonatal rodent models of HIE and stroke confirmed the neuroprotective effect of erythropoietin [45], also when administered as late as starting one week after the occlusion [46]. Subsequently, erythropoietin was tested in near-term animal models, such as the nonhuman primate model of HIE in *Macaca nemestrina*, in combination with hypothermia, producing an improvement of motor and cognitive responses, cerebellar growth, and of the relative risk of developing cerebral palsy at 9 months of age [37]. The same authors have also shown that circulating cytokines and chemokines can be monitored in this model to evaluate the severity of injury and predict the response to therapy with hypothermia and erythropoietin [47]. More recently, experiments using the piglet model of HIE suggest that the administration of earlier melatonin and later erythropoietin (after hypothermia) may provide superior protection in neonates because the two strategies have complementary and time critical actions during the injurious cascade after hypoxia-ischemia [48].

The information that has emerged from these preclinical experiments has been successfully transferred into the clinical setting. Multiple clinical trials have shown that erythropoietin alone or in combination with hypothermia is safe and effective in improving the neurological outcomes in neonates with HIE. Whereas early phase clinical trials have established safety and tolerance, phase 2 trials have demonstrated reduced global and regional brain injury in infants treated with erythropoietin injury [49]. Two ongoing randomized placebo-controlled phase 3 trials have been designed to establish the effects of i.v. erythropoietin on death and disability at 24 months in infants with moderate or severe HIE [50]. However, because randomized and non-randomized trials with erythropoietin have assessed neurodevelopmental disorders in infants with HIE, the importance of including only randomized-controlled studies in systematic reviews and meta-analyses has been recently recognized [51].

### 5.2. Memantine and Topiramate

Another paradigmatic example of an experimental protocol that has developed a promising therapy for HIE regards the glutamate receptor antagonists memantine and topiramate. Whereas memantine is a low affinity NMDA receptor antagonist used in Alzheimer’s disease, topiramate is an AMPA/kainate receptor antagonist and use-dependent sodium channel blocker that has been approved as a therapeutic agent for partial and generalized seizures. Both of them, alone or in combination, proved to be neuroprotective in initial in vivo HIE models [52].

In a study that we performed in collaboration with the “A. Meyer” University Children’s Hospital in Florence, we started by performing OGD experiments in rat organotypic hippocampal slices; incubation of cultures with topiramate and memantine or hypothermia (35 °C or 32 °C) attenuated CA1 damage after 24 h, but the combination of hypothermia with topiramate and memantine improved their protective effects [29]. When tested in vivo in the Vannucci–Rice model, the combination of topiramate or memantine with hypothermia induced a reduction of brain damage that was greater than that produced by drugs or hypothermia alone. Interestingly, memantine produced a superior degree of neuroprotection as compared to topiramate, both in vitro and in vivo, and when used alone at 20 mg/kg in vivo produced a greater reduction in brain damage than observed using topiramate in combination with hypothermia.

Despite our preclinical study showing that memantine provides a superior degree of neuroprotection as compared to topiramate under the same conditions, parallel pilot clinical observations demonstrated the safety and efficacy of topiramate in addition to hypothermia in newborns with HIE and therefore was selected as the therapeutic agent to be added to hypothermia for the completion of the multicenter randomized-controlled NeoNATI trial [53]. Results of this study showed that topiramate plus hypothermia reduced the prevalence of epilepsy in treated neonates but was unable to prevent the combined frequency of mortality or severe neurological disability. More recently, similar trials were able to show that topiramate plus cooling was able to reduce the occurrence of seizures in neonates that achieved therapeutic levels in the first hours after the initiation of the treatment [54].

### 5.3. Cell-Based Treatments

Therapies for treating neonatal HIE based on umbilical cord blood cells, placenta-derived stem cells, and mesenchymal stem cells (MSCs) have recently progressed to early clinical trials [55].

Intranasal delivery of stem cells to target the brain has attracted major interest in the field, as cells can migrate from the nasal cavity to the injured area and exert therapeutic effects, and several research groups have focused on this strategy to treat HIE in neonates [56]. The main mechanism of most cell-based therapies is hypothesized to stem from their bystander effects rather than replacement of lost cells, namely, engraftment and differentiation into neuronal cells [57].

Delivery of stem cells, in particular MSCs, to the brain can attenuate the aberrant inflammation cascade following HIE and provide a more ideal environment for endogenous neurogenesis and repair. Such effects can occur in the subacute phase, thus up to several days after the acute phase of injury, making a cell-based therapy eligible for a delayed intervention [58]. However, timing of application, and definition and standardization of the efficacious doses and the administration route for delivering stem cells to the brain require further optimization to translate the preclinical findings into clinical trials. Thus, the development of novel cell-based therapies will require carefully designed translational studies in both rodents and large animal models to optimize and standardize therapeutic protocols, demonstrating safety and efficacy before instigating complex multi-center clinical trials.

## 6. State of the Art in Italy

In this section we will review the experimental work of some research groups in Italy that have used in vitro and in vivo models of HIE and have been involved in HIE clinical trials in the past few years.

### 6.1. In Vitro Models of HIE in Italy

Experimental models of hypoxia–ischemia in vitro can be used to mimic neonate HIE but also focal or global transient ischemia in adults. There are no particular experimental details in these models that can be recognized as specifically designed or useful for understanding mechanisms or developing new therapies for neonate HIE in humans, except perhaps when selecting tissue from newborn pups or when the effects of drugs are evaluated in combination with hypothermia.

If we bear this in mind, we can recognize various groups in Italy that use in vitro models of hypoxia–ischemia. For example, Marina Pizzi and coworkers in Brescia have performed a number of studies investigating the neuroprotective effects of diverse compounds in primary neuronal cultures exposed to OGD as a model of stroke [59,60]. The same model has also been used by the group of Lucio Annunziato in Naples to explore the role of Na^+^/Ca^2+^ exchangers in cerebral ischemia but also the mechanisms of ischemic preconditioning [61]. Finally, the vulnerability to OGD of primary neuronal cultures derived from Tg2576 Alzheimer mice has been recently examined in the laboratory of Laura Calzà in Bologna [62]. To our knowledge, our laboratory in Florence is the only one in Italy that has used, since 1999, mixed astrocyte–neuronal cultures as a model not only of cerebral ischemia [63], but also of apoptotic and necrotic neuronal death induced by mild or intense NMDA exposure [64,65,66] and of senescence [67].

The group in Naples has considerable experience also with the use of organotypic hippocampal slices exposed to OGD, which have been used in their laboratory to study the neuroprotective effects of the anticonvulsants retigabine and flupirtine [68] and of NCX1 (an Na^+^/Ca^2+^ exchanger) overexpression [69]. This model has been repeatedly used also by our group in Florence essentially for the same purposes as mixed cortical cultures [27,70], but similarly to the strategy of the laboratory of Walter Balduini in Urbino [28], we have specifically used organotypic hippocampal slices exposed to OGD as a first step in the identification of candidate drugs to be used in more complex models of HIE and in infants with HIE in clinical trials [29].

### 6.2. In Vivo Models of HIE in Italy

A few of the laboratories in Italy that we have already mentioned have occasionally used the Rice–Vannucci HIE rat pup model to explore the neuroprotective effects of selected drugs [71] or in search of potential biomarkers for neuroinflammation and neurodegeneration [72]. However, as mentioned in the previous paragraph, our laboratory in Florence and the group in Urbino have designed studies that include both experiments with organotypic hippocampal slices exposed to OGD and parallel, or subsequent experiments performed using the Rice–Vannucci model of HIE with the specific aim of characterizing new drugs to be used in clinical trials. Whereas in our laboratory we concentrated on memantine and topiramate as described in a previous paper [29], Balduini and co-workers have examined in this in vivo model the potential of various drugs, such as simvastatin [73] and, most importantly, melatonin [74], which is now been tested in an Italian study protocol for a randomized-controlled trial [75].

### 6.3. HIE Clinical Trials in Italy

The preclinical studies that we have described in the previous sections have substantially led to two separate clinical trials performed in Italy with Italian Hospital Centers. The first one is the already-mentioned phase II NeoNATI clinical trial (see Section 5.2) with topiramate plus hypothermia directed by the Neonatal Intensive Care Unit of the Children’s Hospitals in Florence and Pisa, which has been completed with encouraging results [53]. The second trial (MELPRO), with melatonin, is a randomized double blind, placebo-controlled trial on 100 neonates with moderate to moderately to severe hypoxic ischemic encephalopathy, which is coordinated by the University Hospital of Ferrara, involves the participation of other five Italian Centers, and is estimated to be completed by December 2022.

## 7. Conclusions

The use of a precise, carefully selected screening cascade of preclinical experimental in vitro and in vivo models can help to improve strategies and develop therapies to protect babies with moderate to severe encephalopathy (Figure 1).

Although cell-based and organotypic models coupled with drug metabolism and pharmacokinetic studies can guide the screening and selection of neuroprotective drugs, in vivo model testing is required to prove efficacy. Nevertheless, there is no single animal model that can fully recapitulate the complexity of HIE in neonates (Table 1).

Rodents model are the most cost-effective models in a drug discovery setting, offering also genetic homogeneity. Large animals, such as piglets, are important not only to further evaluate therapeutic efficacy in gyrencephalic brains, but also for more accurate studies on pharmacokinetics and initial safety pharmacology. Regardless of the model utilized, neuroprotective treatments should be initiated after the completion of the insult to simulate a therapeutic intervention. Given the intrinsic variability of the HIE models and the methodological differences between laboratories, an accurate quantification of the degree of neuroprotection of a given agent can result in variable outcomes. To mitigate this limitation, the NMDA antagonist memantine, in virtue of its known neuroprotective effects, could be used head-to-head with new therapeutics to benchmark efficacy in rodent models of HIE. In summary, a preclinical proof of concept of efficacy of an emerging therapeutic strategy should be obtained in at least two different HIE animal models by independent laboratories before initiating clinical trials.

## Figures and Tables

**Figure 1 biomedicines-10-00937-f001:**
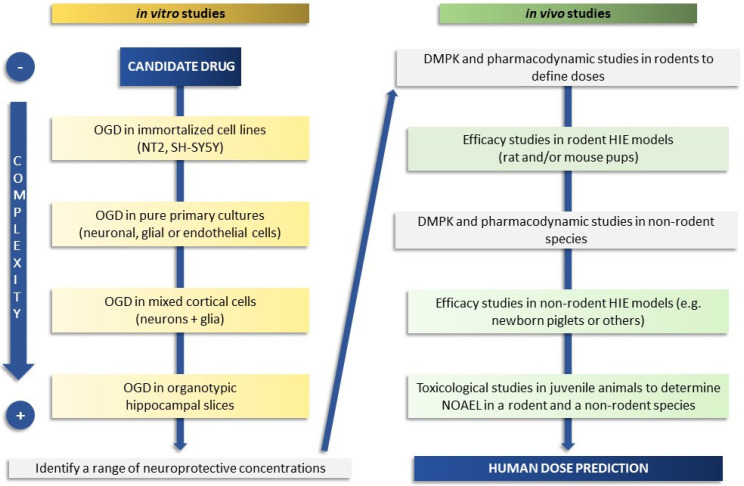
Schematic flow chart of a screening cascade to prove the efficacy of a candidate drug in neonatal HIE. On the left side is depicted a representative test cascade for evaluating the efficacy in in vitro models of increasing complexity (from cell-based to organotypic models) with the aim of defining a range of neuroprotective drug concentrations. Typically, a battery of additional in vitro tests is also used for measurement or prediction of physical properties, drug metabolism, and pharmacokinetic parameters (not fully covered here). Drug metabolism and pharmacokinetic (DMPK) studies, used to understand drug exposure and brain penetration and to define doses, are conducted before progressing to in vivo efficacy studies, shown on the right side. Pharmacodynamic in vivo studies may be required, depending on the mode of action of the candidate drug, before proceeding to efficacy studies. Selected candidate drugs are progressed to studies in rodents (better if two species or different strains and laboratories if using one species such as rat), and subsequently to higher species if feasible, to determine efficacy and the link between target inhibition and neuroprotection. Safety studies in juvenile animals and the determination of the No Observed Adverse Effect Level (NOAEL) in at least two species (rodents and non-rodents) are also required before a candidate drug can be progressed to human studies and the human dose predicted.

**Table 1 biomedicines-10-00937-t001:** Strengths and limitations of the principal in vitro and in vivo models of neonatal HIE. BCAO: bilateral carotid artery occlusion, DIV: days in vitro, E: embryonal day, ICU: intensive care unit, MCAO: monolateral carotid artery occlusion, OGD: oxygen–glucose deprivation, P: post-natal day, PK: pharmacokinetics, UCO: umbilical cord occlusion.

Model	Cell/Tissue/Animal	Exposure	Strengths	Limitations
OGD in vitro				
Immortalized cell lines	NT2, SY5Y	4 h OGD (95% N_2_, 5% CO_2_)	Simple, reproducible	Distant from normal CNS cells
Primary neuronal cultures	7 DIV neurons from E17 rodents	3 h OGD	CNS-like homogeneous cells	No neuron-glia interactions, low response to OGD
Mixed cortical cells	21 DIV astrocytes from P1 mice+ 14 DIV neurons from E17 mice	1 h OGD	Neuron–glia interactions, selective neuronal vulnerability to OGD	Artificial architecture
Organotypic hippocampal slices	14 DIV slices from P8 rats	30 min OGD	CNS-like structural and synaptic organization, CA1 vulnerability to OGD	No vessels
HIE in vivo				
Rice–Vannucci model	P7 rats or mice	MCAO + 2 h 92% N_2_, 8% O_2_	Most convenient, cost-effective, and widely used to study effects of drugs and hypothermia	Lissencephalic brain, variability, mild neurological deficits
Piglet model	P2 piglets	45 min 10% O_2_	Gyrencephalic brain, i.v. drugs + hypothermia, accurate PK	No follow-up on neurological deficits, variability, requires ICU
Intra-uterine models	E29 rabbits	40 min uterine ischemia	Dystonic hypertonia post-natally, complementary to rodents	Limited accessibility
	Fetal sheep	BCAO or UCO	Hypothermia, intrauterine pathophysiology	Maternal/placenta metabolism, cost, and complexity
	Pre-term *Macaca nemestrina*	15 min UCO	Hypothermia, cerebral palsy-like abnormalities	Cost and complexity

## Data Availability

Not applicable.

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
