# Peer review of "Experimental Models for Testing the Efficacy of Pharmacological Treatments for Neonatal Hypoxic-Ischemic Encephalopathy"

_biomedicines, 2022, doi:10.3390/biomedicines10050937_

Round 1
Reviewer 1 Report
The review “Experimental models for testing the efficacy of pharmacological treatments for neonatal hypoxic-ischemic encephalopathy (HIE)…”by E.Landucci et al is timely and useful review for practical medicine allowing reasonable choice of model for pharmacokinetic studies on HIE. Advantages and disadvantages of different in vitro and in vivo models are reviewed with enough details and with the examples of pharmaceuticals used. The consequences of hypoxia-ischemia are associated with oxidative stress and impair of mitochondrial functions. This point and different antioxidant treatments need more detailed discussion in the review. Among great number of different antioxidants only melatonin is mentioned while antioxidant treatment especially by mitochondria-targeted antioxidants, plastoquinone derivatives, proved its efficacy in rat model of neonatal hypoxic-ischemic brain injury. There is one more modern approach – usage of multipatent mesenchimal stem cells (MMCC). It was proved that cocultivation of neuronal cells with MMCC impoves efficacy of the recovery of neuronal cells due to transfer of healthy mitochondria from MMCC induced by Rho GTP-ase Miro1. I presume that stem cells might be not in practice of Italian doctors but the review is written to expand the opportunities of practical medicine.
Author Response
1st Query:
“The consequences of hypoxia-ischemia are associated with oxidative stress and impair of mitochondrial functions. This point and different antioxidant treatments need more detailed discussion in the review. Among great number of different antioxidants only melatonin is mentioned while antioxidant treatment especially by mitochondria-targeted antioxidants, plastoquinone derivatives, proved its efficacy in rat model of neonatal hypoxic-ischemic brain injury.”
Response:
As requested, we have added in the Introduction more detail on the use of mitochondrial-targeted antioxidants.
2nd Query:
“There is one more modern approach – usage of multipatent mesenchimal stem cells (MMCC). It was proved that cocultivation of neuronal cells with MMCC impoves efficacy of the recovery of neuronal cells due to transfer of healthy mitochondria from MMCC induced by Rho GTP-ase Miro1. I presume that stem cells might be not in practice of Italian doctors but the review is written to expand the opportunities of practical medicine.”
Response:
As requested, we have added a whole new subchapter (5.3) on the approaches based on umbilical cord blood cells, placenta-derived stem cells, and mesenchymal stem cells (MSCs), that have been used in models of HIE.
Reviewer 2 Report
The review manuscript titled as “Experimental models for testing the efficacy of pharmacological treatments for neonatal hypoxic-ischemic encephalopathy: a practical approach and state of the art in Italy “ by Landucci et al. provided a very extensive review on cell-based and organotypic models, as well as in vivo models for screening and testing neuroprotective drugs for neonatal hypoxic-ischemic encephalopathy (HIE). The review concluded that there is no single animal model that can fully recapitulate the complexity of HIE in neonates. The review also concluded that a preclinical proof of concept of efficacy of an emerging therapeutic strategy should be obtained in at least two different HIE animal models by independent laboratories before initiating clinical trials. This manuscript was well written and should be of wide interests to most researchers on neuroscience and biomedicines.
Author Response
We thank this reviewer for the positive comments.
Reviewer 3 Report
The authors described cellular and animal models that are used both worldwide and in Italy for important studies of mechanisms and treatment modalities of neonatal hypoxic-ischemic encephalopathy (HIE).
The authors need to describe an influential class of supracellular models like brain organoids in modelling neonatal HIE.
There are examples in the literature. Since they are much more complex than cellular models and reproduce well early stages of human brain development, they are closer to the human than cellular models while retaining relative simplicity in obtaining, low cost and chance to save lives for animals.
Describing models, the authors mentioned generally their face validity, but almost nothing is said about constructive validity: i.e., to what extent the models display similarity with biochemical, molecular and cellular phenotypes of HIE?
What are particular weak spots of models described by the authors, and what advices the authors could give to the readers to help choose the adequate model for HIE research and medical testing?
Moreover, the authors need to add more details about to what extent HIE phenotypes and associated molecular mechanisms are reverted upon the medical treatment of HIE models?
Author Response
1st Query:
“The authors need to describe an influential class of supracellular models like brain organoids in modelling neonatal HIE. There are examples in the literature. Since they are much more complex than cellular models and reproduce well early stages of human brain development, they are closer to the human than cellular models while retaining relative simplicity in obtaining, low cost and chance to save lives for animals.”
Response:
As requested, we have added a whole new subchapter (2.2) on supracellular models and 3D brain organoids.
2nd Query:
“Describing models, the authors mentioned generally their face validity, but almost nothing is said about constructive validity: i.e., to what extent the models display similarity with biochemical, molecular and cellular phenotypes of HIE?”
Response:
We have thoroughly discussed for many models the similarities and difference with the in vivo and human changes that occur after hypoxia and ischemia. We are afraid that further additions may render the manuscript a bit redundant and out of the scope of our review.
3rd Query:
“What are particular weak spots of models described by the authors, and what advices the authors could give to the readers to help choose the adequate model for HIE research and medical testing?”
Response:
In the Table, both strong and weak points of each model are outlined, and in many cases they are also described in the text. As for point 3, we are afraid that further additions may be redundant.
4th Query:
“Moreover, the authors need to add more details about to what extent HIE phenotypes and associated molecular mechanisms are reverted upon the medical treatment of HIE models?”
Response:
We have added a sentence in the Conclusion addressing this point.
Round 2
Reviewer 3 Report
The authors substantially improved the manuscript and I have no further comments.